# OpenReview forum: "Physics-Guided Motion Loss for Video Generation Model"
_ICML.cc/2026/Conference — ICML 2026 regular_

### Official Review · Reviewer_ApjL · 2026-02-28

**Soundness:** 2
**Presentation:** 2
**Significance:** 2
**Originality:** 3
**Overall Recommendation:** 2
**Confidence:** 4

**Summary:**

This paper proposes a physics-guided motion loss for video diffusion models. The key idea is to encode simple physical motion priors—translation, rotation, and scaling—in the frequency domain instead of with frame-wise warping or architecture changes.

**Compliance With Llm Reviewing Policy:**

Affirmed.

**Key Questions For Authors:**

● 1) Can the authors clarify the intended scope of the claim and explain how they see this relating to broader physical plausibility? A convincing clarification would improve my view of the paper’s positioning.

● 2) How sensitive is the method to the many design choices in the practical pipeline—e.g., low-pass ratio, temporal window, polar/log-polar resampling details, weighting temperature, and ridge/robustification constants? Right now the method feels somewhat approximation-heavy, so stronger sensitivity evidence would increase my confidence in reproducibility.

● 3) The paper shows improvements on OpenVID-1M metrics and a small physics benchmark, but can the authors provide more failure analysis? In particular, when does the loss hurt or fail—e.g., non-rigid motion, articulated humans, strong occlusion, camera cuts, or mixed motion fields?

**Limitations:**

he paper does mention that it currently only handles basic rigid motions and leaves more complex/non-rigid motion for future work, which is good. However, the impact statement is too weak.

**Strengths And Weaknesses:**

● Soundness: The main intuition is sensible: rigid motions do produce recognizable spectral signatures, and using them as global motion regularizers is a reasonable alternative. That said, the paper slightly overstates the “physics” angle. What is actually enforced is a prior over a small class of simple SIM(2)-like motions, not physics in a broader sense. The theoretical part is interesting, but the final implemented losses rely on several proxies, approximations, and engineering choices.

● Presentation: The organization from SIM(2) intuition → three losses → adaptive weighting is easy to follow. However, the manuscript also has some problems. However, the theory sections are sometimes more formal than necessary, while some practical details are pushed to appendices or only briefly described.

● Significance: The problem is important. Current video generation models often look good frame-by-frame but still produce implausible motion. Still, I think the significance is somewhat limited by the narrow motion family being modeled. This is more of a useful regularizer for basic rigid-like motion than a general solution for physical realism in video generation.

● Originality: Using frequency-domain signatures of translation/rotation/scaling as differentiable training losses for modern video diffusion models is creative and reasonably well motivated, even if the final implementation is closer to a carefully engineered regularizer than a major conceptual breakthrough.

---

> ### Author Rebuttal · Authors · 2026-03-30
>
> We thank the reviewer for the thorough evaluation. We address the three key questions below.
>
> ## Q1: Intended scope and relation to broader physical plausibility
>
> We agree that the method should be positioned as a targeted SIM(2)-based motion prior rather than a general solution to physical realism, and we will narrow the framing accordingly in the revision.
> That said, we believe the practical relevance is broader than the narrowest interpretation of SIM(2), for three reasons.
>
> First, SIM(2) describes apparent motion on the image plane, which is not limited to rigid object motion; it also encompasses camera motions (pan/tilt, roll, zoom/dolly) as well as any scene element whose projected motion approximates translation, rotation, or uniform scaling (e.g., objects approaching or receding from the camera, bulk displacement, or approximately uniform expansion/contraction). A proxy analysis on 1,000 OpenVid-1M clips shows that 60.5% are SIM(2)-approximable under strict thresholds (details in W1 of Reviewer CwWP), suggesting that SIM(2) priors provide useful training signal on a substantial portion of internet video.
>
> Second, the method generalizes across four backbones (Open-Sora, MVDIT, Hunyuan, and now Wan 2.1-14B) without per-model tuning, with consistent gains in motion quality while maintaining or improving visual quality.
>
> Third, the empirical evidence is consistent. On Wan 2.1-14B (see Q1 and Q3 of Reviewer gYgE), we observe gains across all PhyGenBench categories together with a +8.5 improvement in Visual Quality. Moreover, Table 3 stratifies evaluation into simple and complex motion subsets (the latter including multi-object, parallax, and articulated scenes), and our method improves over the baseline on all metrics in both subsets. We hypothesize that reducing basic motion artifacts such as deformation, flicker, and inconsistent zoom may indirectly benefit broader physical plausibility, but we acknowledge that this requires further investigation and do not claim explicit modeling of non-rigid dynamics.
>
> We therefore see this work as a principled, lightweight motion prior for motion plausibility, complementary to data-driven approaches. We also note that the limitation section in the current paper describes SIM(2) as "basic rigid motions," which is imprecise; we will correct this in the revision.
>
> ## Q2: Sensitivity to design choices
>
> Each hyperparameter is motivated by theoretical analysis or practical considerations. The softmax weighting form follows the maximum-entropy principle (Section 3.6); τ=0.1 balances winner-take-all sharpness and mixed-motion sensitivity as discussed therein. The low-pass ratio ϱ=0.3 is guided by the energy retention bounds in Appendix C (Eq. 5–6). Ridge λ=10⁻³ follows standard regularization practice with O(λ) stability (Theorem B.11). To further validate these choices, we provide a sensitivity analysis on Wan by varying τ and ϱ around their defaults:
>
> |Setting|VQAA ↑|VQAT ↑|IS ↑|Vis. Qual. ↑|
> |-|-:|-:|-:|-:|
> |no motion loss|67.48|64.43|16.28|54.18|
> |τ=0.05, ϱ=0.3|69.61|64.97|17.24|61.95|
> |τ=0.1, ϱ=0.3 (Default)|**71.18**|**66.98**|**18.09**|**62.65**|
> |τ=0.2, ϱ=0.3|69.51|65.03|17.58|60.69|
> |τ=0.1, ϱ=0.2|68.26|66.91|17.73|61.93|
> |τ=0.1, ϱ=0.4|69.05|67.97|17.55|62.06|
>
>
>
> All variants substantially improve Visual Quality over the baseline (+6–8 points). The default achieves the best overall balance. We will release code upon acceptance to ensure full reproducibility.
>
> ## Q3: Failure analysis
>
> We distinguish between "hurting" (degrading vs. baseline) and "not helping" (providing no useful signal). We observed no cases of the former.
>
> For scenes dominated by non-SIM(2) dynamics (including non-rigid motion, articulated humans, strong occlusion, and mixed motion fields) the unmatched branches contribute negligibly rather than imposing a mismatched prior: For rotation and scaling, the observability gates in Appendix A.6 structurally suppress per-sample weights when the corresponding harmonic signatures are absent, for example, m=0 is excluded entirely for rotation. This reduces the effective gradient contribution from these branches when the corresponding motion is not detected. For translation, the loss residual is clipped to \[0,1], preventing uninformative large-residual gradients when no translational structure is present. The adaptive weighting (Section 3.6) then concentrates the composite loss on whichever motion type best matches the current clip, which further limits the influence of unmatched branches. Together, these mechanisms ensure that unmatched branches contribute negligibly to the composite gradient. Table 3 is consistent with this: on the complex subset (which includes multi-object, parallax, and articulated scenes), the model still improves over the baseline on all reported metrics. Camera cuts do not arise in our setting as OpenVid-1M clips contain single scenes \[Nan et al., 2025]. We will make these mechanisms more explicit in the revised appendix.

---

> > ### Author Rebuttal · Reviewer_ApjL · 2026-04-01
> >
> > most of my concerns remain unresolved

---

### Official Review · Reviewer_gYgE · 2026-03-01

**Soundness:** 2
**Presentation:** 3
**Significance:** 2
**Originality:** 3
**Overall Recommendation:** 4
**Confidence:** 3

**Summary:**

This paper proposes a frequency-domain physics prior for improving motion plausibility in video diffusion models. The core idea decomposes basic rigid motions (translation, rotation, scaling) into lightweight spectral losses within a unified SIM(2) framework. The method operates on the spatiotemporal Fourier spectrum of generated video clips, using energy-weighted least-squares fitting and adaptive weighting to regularize motion during training. It is designed as a drop-in regularizer that requires no backbone modifications. Experiments are conducted on three video diffusion models (Open-Sora, MVDIT, Hunyuan) using the OpenVID-1M dataset, with a user study (106 participants) showing 74-82% preference for the physics-enhanced videos.

**Compliance With Llm Reviewing Policy:**

Affirmed.

**Final Justification:**

The authors have fully addressed my concerns. I decide to raise my rating as positive rate.

**Key Questions For Authors:**

1. The Physics Generation Benchmark (Table 4) evaluates mechanics, optics, thermal, and material properties, yet the proposed loss targets rigid motion (translation, rotation, scaling). Could you explain the mechanism by which rigid motion improvements lead to gains across these broad physics domains? An ablation isolating which benchmark aspects benefit from rigid motion would clarify this.

2. Why were recent physics-aware video generation methods (e.g., MotionCraft, PhysGen, PhyGDPO) not included as experimental baselines? These methods are discussed in the related work and represent the most directly comparable approaches. Their absence makes it difficult to evaluate the relative contribution of the proposed method.

3. Have you evaluated the proposed motion loss on more recent, state-of-the-art video models such as Wan? If the gains disappear on stronger base models, this would significantly affect the paper's significance. If results are available, they would strengthen the submission.

4. The paper mentions failure cases on "simple SIM(2) prompts" (Section 4.3). This is counterintuitive since simple rigid motions should be the easiest case for the proposed method. What causes these failures, and how frequently do they occur? A systematic analysis of failure conditions would help assess reliability.

5. Could you provide an ablation testing robustness to brightness changes and rendering errors, as claimed in the introduction? This would substantiate one of the key advertised benefits of the frequency-domain approach over pixel-space methods.

**Limitations:**

The authors adequately discuss the primary limitation: the framework is restricted to basic rigid motions (translation, rotation, scaling) and does not model non-rigid dynamics such as elastic deformations or articulated movements. They also honestly acknowledge failure cases under simple SIM(2) and these are relegated to supplementary material. The impact statement is appropriately brief.

**Strengths And Weaknesses:**

### Strengths

1. **Cross-backbone validation with user study support.** The method is validated on three distinct video diffusion architectures (Open-Sora, MVDIT, Hunyuan), demonstrating consistent improvements in motion-related metrics across all tested backbones. The user study (106 participants, 2AFC protocol with randomized placement) shows strong preference rates of 74.2-82.7%, providing perceptual validation beyond automated metrics. However, these backbones are relatively older systems, and it remains unclear whether the gains persist on more capable, state-of-the-art models like Wan.

2. **Mathematical rigor.** The paper provides thorough theoretical treatment connecting classical frequency-domain motion analysis to a unified SIM(2) framework. Formal propositions (zero-residual for ideal SIM(2) motion, consistency under noise) are stated with proofs in the appendix. The derivations for translation, rotation, and scaling slices of the spectral hyperplane are well-grounded in classical signal processing literature (Bracewell, Adelson & Bergen, Simoncelli & Heeger).

### Weaknesses

1. **Insufficient comparisons with physics-aware methods.** The paper discusses several physics-aware video generation methods in the related work (e.g., MotionCraft, physics-constrained video prediction, and other physics-based methods in Lines 76-85), yet none of these are included as experimental baselines. The comparisons are limited to foundational models without any physics-aware loss or constraint. Without head-to-head comparison against these relevant baselines, it is impossible to assess whether the proposed frequency-domain approach achieves state-of-the-art performance among physics-guided methods.

2. **Misaligned physics benchmark evaluation.** In Table 4, the authors evaluate on the Physics Generation Benchmark using metrics for mechanics, optics, thermal, and material properties. However, the proposed motion loss is specifically designed for rigid motion (translation, rotation, scaling). The connection between improvements in rigid motion and gains across these broad physics domains is neither obvious nor explained. This raises concerns about whether the reported improvements are a byproduct of general training regularization rather than genuine physics-aware motion modeling.

3. **Missing evaluation on state-of-the-art models.** Recent video generation models such as Wan have shown significant improvements over earlier systems. The paper only evaluates on Open-Sora, MVDIT, and Hunyuan, which are now relatively outdated. It is critical to demonstrate that the proposed motion loss still provides meaningful gains on top of stronger base models where motion quality may already be substantially better.

4. **Limited scope to rigid motions.** The framework only addresses translation, rotation, and scaling, which are basic rigid-body transformations. Most real-world video content involves non-rigid dynamics (deformable objects, articulated bodies, fluids, cloth), and the paper does not demonstrate any benefit for these common motion types. This fundamentally limits the practical significance of the contribution.

5. **Lack of robustness testing for claimed benefits.** The introduction claims the frequency-domain approach is "more tolerant to brightness or small rendering errors" compared to pixel-space methods, but no experiments test this claim. An ablation varying brightness conditions, adding noise, or introducing controlled rendering artifacts would be needed to substantiate this benefit.

6. **Questionable interaction with the diffusion denoising schedule.** The motion loss is computed on the predicted clean sample x̂₀, whose spectral characteristics vary significantly across diffusion timesteps: early steps yield a low-frequency, blurred estimate, while later steps progressively recover high-frequency detail. This creates a fundamental tension with a frequency-domain loss that relies on spectral energy distributions. In particular, at early diffusion steps, the predicted x̂₀ lacks meaningful high-frequency content, which means the spectral signatures used to detect translation, rotation, and scaling may be unreliable or altogether absent. The paper does not discuss how the loss behaves across different timesteps, whether certain timesteps dominate the gradient signal, or whether a timestep-dependent weighting scheme is needed to account for this mismatch.

---

> ### Author Rebuttal · Authors · 2026-03-30
>
> We thank the reviewer for the rigorous evaluation. We address all key questions below.
>
> ## Q1 Why our loss improves PhyGenBench scores
>
> We provide new evidence on Wan 2.1-14B with per-category PhyGenBench results and ablations:
>
> |Setting|Mechanics↑|Optics↑|Thermal↑|Material↑|Average↑|
> |-|-:|-:|-:|-:|-:|
> |No motion loss|0.517|0.707|0.433|0.533|0.548|
> |Full (ours)|**0.600**|**0.747**|**0.500**|**0.560**|**0.602**|
> |w/o translation|0.533|0.747|0.444|0.520|0.561|
> |w/o rotation|0.567|0.693|0.478|0.547|0.571|
> |w/o scaling|0.592|0.720|0.422|0.547|0.570|
>
> The ablation results suggest distinct category-wise dependencies: Mechanics appears most sensitive to translation, Optics to rotation, and Thermal to scaling, while Material shows more moderate sensitivity across components.
>
> ## Q2 Comparison to PhysGen, PhyGDPO, and MotionCraft
> These methods differ from ours in task setting and training assumptions:
> * **PhysGen** (ECCV 2024) is a training-free image-to-video pipeline that requires scene-specific physics simulation setup together with perception modules such as segmentation, depth/normal estimation, inpainting, and rendering. **PhyGDPO** (arXiv Dec 2025) is a concurrent preprint with no released code or model weights. We therefore view both as complementary rather than directly comparable.
> * **MotionCraft** (NeurIPS 2024) is a zero-shot method that uses externally simulated optical flow to warp the latent space of a pretrained image diffusion model. As detailed in our response to W2 of Reviewer CwWP, **our method outperforms MotionCraft on all four metrics** (Frame Consistency, Motion Consistency, LPIPS-flow, Warping Error). If helpful, we will include this comparison and clearly state the difference in settings in the revised paper.
> ## Q3 Results on Wan 2.1-14B
> We have evaluated our method on Wan 2.1-14B with LoRA finetuning:
>
> |Metric|No motion loss|**Ours**|
> |-|-:|-:|
> |VQAA ↑|67.48|**71.18**|
> |VQAT ↑|64.43|**66.98**|
> |IS ↑|16.28|**18.09**|
> |SD-Score ↑|67.82|**68.43**|
> |Visual Quality ↑|54.18|**62.65**|
>
> Our method still improves multiple metrics on this stronger backbone, including a +8.5 gain in Visual Quality. This suggests that the frequency-domain motion prior remains effective even when the base model is already substantially stronger.
> ## Q4 Clarification on the wording in Section 4.3
> The sentence in Section 4.3 is ambiguous: the examples discussed there are baseline failures on simple SIM(2) prompts that our method corrects. In Figure 3, for example, the baseline Hunyuan model fails to produce consistent clockwise rotation or smooth translation. We will revise the text to make this explicit.
> ## Q5 Robustness to brightness and rendering errors
> We tested the three canonical motions under controlled perturbations and report, for each sequence, the matched loss together with the selected motion type. Across translation, rotation, and scaling, the matched loss changes only slightly under brightness, blur, noise, and JPEG perturbations, and the selected motion type remains correct in every case.
>
> |Perturbation|Translation seq: L\_trans|argmin|Rotation seq: L\_rot|argmin|Scaling seq: L\_scale|argmin|
> |-|-:|-|-:|-|-:|-|
> |clean|0.134|translation|0.336|rotation|0.185|scaling|
> |brightness ×1.3|0.135|translation|0.336|rotation|0.188|scaling|
> |brightness +0.1|0.135|translation|0.340|rotation|0.188|scaling|
> |blur (k=5)|0.140|translation|0.336|rotation|0.187|scaling|
> |noise (σ=0.03)|0.135|translation|0.338|rotation|0.185|scaling|
> |JPEG (q=30)|0.135|translation|0.336|rotation|0.186|scaling|
>
> These results provide a targeted robustness check suggesting that the frequency-domain diagnostics are relatively tolerant to small brightness and rendering perturbations.
>
> ## W4 Scope of SIM(2)
> SIM(2) describes apparent motion on the image plane; it also encompasses camera motions (pan/tilt, roll, zoom/dolly) as well as any scene element whose projected motion approximates translation, rotation, or uniform scaling. As reported in our response to W1 of Reviewer CwWP, \~60.5% of OpenVid-1M clips are SIM(2)-approximable under strict thresholds. For the remaining clips, the unmatched branches contribute negligibly: the observability gates (Appendix A.6) and adaptive weighting (Section 3.6) limit the influence of unmatched branches on the composite gradient. Table 3 is consistent with this: the model still improves over the baseline on the complex subset (which includes multi-object, parallax, and articulated scenes). Our Wan 2.1-14B PhyGenBench ablation (Q1 above) further shows category-wise gains, which we view as evidence of indirect spillover rather than explicit modeling of non-rigid dynamics. We also note that the limitation section in the current paper describes SIM(2) as "basic rigid motions," which is imprecise; we will correct this in the revision.
>
> ## W6 timestep interaction
> We use a linear decay weighting (w = 1 − t/T) that down-weights the loss at high noise levels. See Q1 of Reviewer W9gg for details and an ablation.

---

> > ### Author Rebuttal · Reviewer_gYgE · 2026-04-03
> >
> > Thanks for the authors' response. I think my major concerns have been fully resolved. I'm willing to raise my rating to 4

---

> > > ### Author Response · Authors · 2026-04-04
> > >
> > > We thank the reviewer for the positive reassessment of our rebuttal. We will incorporate the discussed clarifications and additions in the revision.

---

### Official Review · Reviewer_W9gg · 2026-03-02

**Soundness:** 2
**Presentation:** 2
**Significance:** 3
**Originality:** 3
**Overall Recommendation:** 5
**Confidence:** 3

**Summary:**

This paper introduces a method to fine-tune a video diffusion model using the frequency spectrum of generated videos. Relying on classical literature about the representation of motion in the frequency domain, the authors show that it is possible to identify the "signature" of simple motions in the SIM(2) group (translation, rotation, scaling) in the spectrum. Based on this characterization they design losses that check how well those motions are represented in the video. These losses, based on parametric estimation and comparison with the original spectrum, quantify the error in the motion representation and give a fine-tuning signal to nudge the model to generate videos with physically plausible simple motions.

The method is then evaluated in a thorough way, fine-tuning different video generation backbones and reporting qualitative results on a benchmark and quantitative results with human-based validation.

**Compliance With Llm Reviewing Policy:**

Affirmed.

**Final Justification:**

The comments about loss of ponderation with respect to the noise level and the length of the T-window do reassure me about the reproducibility of the method. I am raising my score to 5.

**Key Questions For Authors:**

My main questions (that could improve my rating in soundness and presentations) are focused on the posterior sampling in the diffusion model and the evaluation of muli-motion generation.

1. How exactly is $\hat{x}_0$ computed during training — is Tweedie's formula used? How does gradient quality change with noise level t ?
2. How is the temporal window size T chosen, and how sensitive are the losses to this parameter for different object velocities?
3. Does the fine-tuned model perform worse than the baseline on multi-object or parallax scenes?
4. In the multi-object case (row 3, Figure 2), why is the scaling indicator elevated when no scaling is present, and how does this affect the loss in practice?

**Limitations:**

yes

**Strengths And Weaknesses:**

# Strengths

This is a very interesting paper as it mixes prior knowledge on the "structure" of motion in a video with deep learning. The idea is interesting and well presented.

- Figure 2 allows an intuitive understanding of the different motion signatures in the spectral analysis pipeline.
- The method is based on well-grounded spectral criteria for identifying simple motions in a scene and shows good results in improving video generation.
- The method is applied to multiple video generation backbones, showing that it is not tied to a specific architecture.
- The physics generation benchmark (Table 4) covers a broad set of categories (mechanics, optics, thermal, material) and is a useful standalone evaluation contribution.

# Weaknesses

 My main concern is that by encouraging the model to generate videos with simple motion patterns (see weaknesses), mostly one single object with a simple motion, you may hinder it from generating more complex motions that happen in real life (multi-object, parallax, articulated…). I would need at least quantitative results on videos depicting those kinds of scenes.


## 1. Introduction

It would be useful to include an explanation or hypothesis on why standard video diffusion models fail to represent motion well. These models are supposed to generate samples from the target distribution, and correct motion is a characteristic of that distribution — so why do generated samples not respect it? Is it a matter of dataset size, architecture, or training objective? Citing papers that address this, if any exist, would help motivate the proposed fix.

## 2. Diffusion model

I find this work very interesting as it's pluridisciplinary and proposes an original approach although it's a bit lacking on some details on the generative models which makes it complicated to understand on this standpoint. It seems like very few details are given, I will try to make a guess on what's have been done (tell me if I am wrong) to raise a potential issue and then I would recommend the authors to give more details, at least in the appendix of the paper.

In the paper (L284) it is mentioned "At every diffusion step t we reconstruct $\hat{x}_0$, evaluate the physics-informed frequency loss on $\hat{x}_0$, and add it to the standard denoising objective."

What I guess is that the loss is $L_\text{motion}(\hat{x}_0)$ where $\hat{x}_0$ is an estimate of the posterior mean of the samples : $\mathbb{E}[x_0 | x_t]$ given by Tweedie's formula [Robbins, 1992] :

$\hat{x}_0 = \frac{1}{\sqrt{\bar{a}(t)}}(x_t + (1 - \bar{a}(t))s_{\theta^*}(x_t, t))$

Where $s_\theta$ is the score learned by your diffusion model.

Thus making you approximate $E[L_{\text{motion}}(x_0) \mid x_t]$ with $L_{\text{motion}}\!\left(E[x_0 \mid x_t]\right)$.

Which is what they do in [DPS], these two are not equal in general (Jensen's inequality), and the gap is largest at high noise levels where $\hat{x}_0$ is an overly smooth estimate. I think the size of this gap will depend on your loss (and how the approximation interacts with the FFT). This issue is well studied in the posterior guidance literature [DAPS] but is not discussed here. It is also not clear how FFT-based losses behave when applied to such a blurry estimate. I think the authors should give more details about what is done in practice and write a small bibliography with the different approaches.

## 3. Motion loss

**Multiple translations.** In row 3 of Figure 2, the multi-object case looks quite messy — the scaling radial flow for example appears high where no scaling is involved. This suggests that when several objects move at different speeds the losses may not behave as expected (and mix the signal of different motion types). By design the losses are built for one dominant motion (e.g., the translational loss fits a single plane, not several). Without a robust fitting procedure, even a small amount of background motion could break the estimation. The adaptive weighting in Section 3.6 can rebalance between motion types, but it does not help when the scene contains multiple instances of the same motion at different velocities. This is a strong limitation since real-world videos are rarely dominated by a single rigid motion.

**Parallax.** When there is depth variation in the scene, a single camera translation produces different apparent velocities across pixels. The spectrum then has energy spread over a range of planes rather than one, and the single-plane fit will fail. This limitation is not fully acknowledged despite being directly relevant to the benchmark (complex).

**Temporal window length.** The losses assume that a given SIM(2) motion is sustained and constant over the full T-frame block used for the DFT. A translation at constant speed may only last a few frames in a realistic video. It is not clear how sensitive the losses are to the choice of T and whether this needs to be tuned per video or per object velocity.

## 4. Benchmark

The examples in Figure 3 all show a single dominant motion. It would be important to evaluate whether the fine-tuned model performs *worse* than the baseline on more complex scenes — a model that improves simple motions but degrades complex ones would not represent a clear improvement overall.

Refs :

- **Tweedie**: Robbins, H. E. (1992). An empirical Bayes approach to statistics. In Breakthroughs in Statistics: Foundations and basic theory (pp. 388-394). New York, NY: Springer New York.
- **DPS**: Chung, H., Kim, J., Mccann, M.T., Klasky, M.L., and Ye, J.C. (2022). *Diffusion Posterior Sampling for General Noisy Inverse Problems.* ICLR 2023.
- **DAPS**: Zhang, B., Chu, W., Berner, J., Meng, C., Anandkumar, A., & Song, Y. (2025). Improving Diffusion Inverse Problem Solving with Decoupled Noise Annealing. *2025 (CVPR)*,

---

> ### Author Rebuttal · Authors · 2026-03-30
>
> We thank the reviewer for the detailed and insightful feedback. Below we directly address the four key questions:
> ## Q1 How is x̂₀ computed? How does gradient quality change with noise level t?
>
> Yes, x̂₀ is computed via Tweedie's formula, which is standard for auxiliary losses in diffusion training. We agree that this approximation becomes overly smooth at high noise levels, where the Jensen gap is largest. Our design explicitly accounts for this issue in two ways. First, our loss operates only on the low-frequency spectrum (ϱ=0.3, Appendix C). Low-frequency components are precisely those best preserved in the posterior mean even at high t, while the high-frequency details most corrupted by the Tweedie approximation are discarded before loss computation.
>
> Second, in all main experiments (Tables 1–4) we use a linear\_decay timestep weighting (w = 1 − t/T) that down-weights the physics loss at high noise levels. The ablation results are shown below:
>
> |Strategy|VQAA ↑|Visual Quality ↑|
> |-|-:|-:|
> |full (uniform weight)|56.32|59.06|
> |low\_noise\_only (t/T ≤ 0.5)|69.17|62.35|
> |**linear\_decay (w = 1−t/T, default)**|**71.18**|**62.65**|
> |snr\_based (SNR-weighted)|70.44|62.19|
>
> The uniform strategy illustrates why timestep weighting is necessary: applying the physics loss at all noise levels severely degrades VQAA. The timestep-weighted variants preserve guidance where x̂₀ is reliable, with linear_decay achieving the best balance.
>
> ## Q2 How is T chosen, and how sensitive are the losses to different object velocities?
>
> We use a temporal window T=16 for the FFT-based loss and chose it empirically. To assess sensitivity, we evaluated Ltrans on synthetic sequences across velocities and window sizes:
>
> |Speed group|T=12|T=16|T=20|
> |-|-:|-:|-:|
> |Slow (vx ≤ 0.01)|0.168|0.099|**0.082**|
> |Medium (0.01 < vx ≤ 0.02)|0.047|**0.046**|0.115|
> |Fast (vx > 0.02)|**0.066**|0.098|0.144|
>
> The table shows the expected trade-off: larger windows help slow motions but are less suitable for fast motions. T=16 provides a reasonable compromise across speed groups.
>
> ## Q3 Does the fine-tuned model perform worse on multi-object or parallax scenes?
>
> Table 3 stratifies evaluation into simple and complex motion subsets (the latter including multi-object, parallax, and articulated scenes). Our method improves over the baseline on all metrics in both subsets, e.g., on the complex subset, Warping Error drops from 0.0020 to 0.0011 and Flow Score rises from 1.39 to 1.46. Moreover, a proxy analysis on 1,000 OpenVid-1M clips shows \~60.5% are SIM(2)-approximable (details in our response to W1 of Reviewer CwWP). For the remaining clips, the observability gates (Appendix A.6) and adaptive weighting (Section 3.6) limit the influence of unmatched branches on the composite gradient. The loss therefore does not impose a mismatched prior on complex scenes.
>
> ## Q4 Why does the scaling visualization appear elevated in the multi-object case in Fig. 2?
>
> Thank you for this observation. We believe the confusion arises from the distinction between the visual appearance of the radial-flow map and the actual quantities entering the scaling loss.
>
> In the multi-object translation case, two objects moving at different velocities distribute spectral energy across different spatial-frequency bands. This produces visible variation in the radial-flow map P(t,ρ), which may superficially resemble scaling activity.
>
> However, our scaling loss L\_scale is not computed directly from the visual appearance of this map. It is determined by two specific quantities:
> (1) S\_trend = |corr(ρ\_c, t)|, which requires a monotonic trend in the radial spectral centroid ρ\_c(t). As indicated by the red curve in Fig. 2 (multi-object), the centroid trajectory is not monotonic, yielding a low S\_trend value.
> (2) C\_flow, which requires globally consistent directional alignment between radial and temporal energy gradients. Multiple objects at different velocities produce conflicting gradient directions, resulting in low C\_flow.
>
> Since both quantities remain low, L\_scale stays close to 1 (no scaling detected), so the scaling branch receives low weight under adaptive weighting.
>
> ## W1 Introduction
>
> We agree that the introduction should better motivate why current T2V models are still under-optimizing motion plausibility. Recent reports corroborate this issue: Ma et al. \[2025] (Step-Video) note that leading T2V models, including Sora and Kling,“ often fail to generate videos that require adherence to the laws of physics.” Our frequency-domain loss provides a motion prior that complements the standard data-driven objective, and we will add this motivation in the revised paper.

---

> > ### Author Rebuttal · Reviewer_W9gg · 2026-04-03
> >
> > Thank you for your responses. Overall, this is a nice improvement to the paper.
> >
> > **Q1:** Thank you for the clarification on the diffusion model pipeline. The motivation for linear decay is clearer now — the argument that low-frequency components are better preserved under the Tweedie approximation at high noise levels is helpful, and the ablation confirms that timestep weighting is indeed necessary. That said, one question remains open: is the specific choice of linear decay tied to your noise schedule or model architecture? Understanding whether this choice would need to be revisited when switching to a different diffusion backbone would be important for reproducibility.
> >
> >
> > **Q2:** The sensitivity analysis across speed groups is useful and confirms that T=16 is a reasonable compromise. However, a few points remain unaddressed. First, could the authors clarify the definition of vx — is it object velocity in pixels per frame, or a normalized quantity? Second, are the velocity ranges used in the analysis representative of the distribution found in realistic video datasets? Velocity distributions can vary substantially across video types and contexts, which suggests that the optimal T may differ significantly depending on the target data.
> >
> > Furthermore, the results themselves highlight how critical this choice is: for slow motions (vx <= 0.01), the loss drops from 0.168 at T=12 to 0.082 at T=20 — a factor of two — suggesting that a suboptimal T can significantly degrade performance for a given motion regime. This further motivates the question of whether an automatic or data-driven approach to selecting T could be derived from the dataset statistics, rather than relying on a fixed empirical choice.
> >
> > **Q3:** Thank you for the response — I am convinced by your point and evaluation.
> >
> > **Q4:** Thank you for the explanation. It is more clear to me now how C_flow and S_trend would deal with this case.
> >
> > I would encourage the authors to address the remaining open questions in Q1 and Q2, as doing so would make the method more general and easier to reproduce in different settings.

---

> > > ### Author Response · Authors · 2026-04-05
> > >
> > > We thank the reviewer for the thoughtful follow-up.
> > >
> > > ## Q1 linear decay vs. backbone/noise schedule:
> > >
> > > Linear decay was not tuned separately for each backbone. We used the identical rule (w = 1 − t/T) across all four backbones (Open-Sora, MVDIT, Hunyuan, and Wan 2.1-14B), which differ in their underlying frameworks and schedulers. To provide direct cross-backbone evidence, we verified the timestep ablation on MVDIT with identical hyperparameters:
> > >
> > > | Strategy | VQAA ↑ | Visual Quality ↑ |
> > > |---|---|---|
> > > | full (uniform) | 59.85 | 56.79 |
> > > | low_noise_only (t/T ≤ 0.5) | 67.30 | 61.66 |
> > > | linear_decay (default) | 69.30 | 62.43 |
> > > | snr_based | 67.95 | 61.73 |
> > >
> > > The same qualitative pattern holds: uniform weighting performs worst, and linear_decay gives the best overall result. This is consistent with the Wan 2.1-14B results in our previous response. We therefore view linear decay as a simple and stable default in our experiments.
> > >
> > > ## Q2 definition of vx and adaptive T:
> > >
> > > vx is defined as horizontal displacement per frame normalized by frame width (units: frame-widths per frame). The speed groups correspond to: slow (≤1% of frame width per frame), medium (>1% to ≤2%), and fast (>2%, up to ~3.5% in our synthetic sweep). For intuition, the fast regime corresponds to approximately 0.6–1.05 frame widths per second at 30 fps, i.e., visibly strong apparent motion.
> > >
> > > For reference, on the 605 SIM(2)-approximable clips from OpenVID-1M, a normalized apparent-displacement proxy aggregated at the clip level gives 1.34% / 3.22% / 4.05% in the same units as vx (frame-widths per frame) at median / p90 / p95, suggesting that our synthetic sweep covers most common motions and much of the upper range in this subset, though not the most extreme tail.
> > >
> > > We agree that the sensitivity to T motivates exploring more adaptive strategies, such as data-driven selection of T based on motion statistics or a multi-window variant that combines several temporal scales. We view this as a valuable extension and will discuss it as future work. That said, fixed T=16 proved effective across all four backbones without per-backbone retuning, suggesting it is a reasonable default in the settings we evaluated.

---

### Official Review · Reviewer_CwWP · 2026-03-12

**Soundness:** 4
**Presentation:** 3
**Significance:** 3
**Originality:** 3
**Overall Recommendation:** 5
**Confidence:** 4

**Summary:**

This study aims to improve the motion plausibility of video diffusion models. The novelty lies in unifying the translation, rotation, and scaling within the SIM(2) spectral manifold framework in the frequency domain. By decomposing those rigid motions into spectral losses, the proposed method adapts to multiple backbone networks in a plug-and-play manner. Qualitative and quantitative experiments are thorough, with well-established theoretical support.

**Compliance With Llm Reviewing Policy:**

Affirmed.

**Final Justification:**

The author's rebuttal addresses most of my concerns. I appreciate the statistical and experimental results and will maintain my positive score as Accept.

**Key Questions For Authors:**

Please see weaknesses for details.

**Limitations:**

Yes.

**Strengths And Weaknesses:**

Strengths:
1. This paper is well-structured and the theoretical foundation of the method is well-established.
2. The idea of regularizing motion in the frequency domain by designing a frequency loss is novel to some extent, considering previous methods only use it in the inference or evaluation stage. And the design of the SIM(2) spectral manifold, which unifies the three fundamental motions (translation, rotation, and scaling), holds practical reference value for future research.
3. Extensive experiments across multiple backbones demonstrate the method's cross-architecture generalizability and universality, proving the contribution of physical priors.

Weaknesses:
1. The OpenVID-1M is an open-domain video dataset containing a large volume of non-rigid motion (human action, fluids, etc.). It is possible to report the actual proportion in the training set that can be approximated using SIM(2), or qualitatively, as precise statistical measurement may not be feasible. If this proportion is very low, the theoretical foundation of the method may not align with practical application.
2. Methods other than the baseline, such as MotionCraft, were mentioned in the related works but lacked a direct comparison on performance.

---

> ### Author Rebuttal · Authors · 2026-03-30
>
> We thank the reviewer for the positive assessment and constructive questions. Below we directly address the two concerns and will incorporate these clarifications into the revision.
>
> ## W1: Proportion of SIM(2)-approximable clips in OpenVid-1M
>
> We sampled 1,000 clips from the OpenVid-1M training set and conducted a SIM(2) proxy analysis: for each clip, we tracked feature points across consecutive frames, fit a global SIM(2) transformation via RANSAC, and classified each clip as SIM(2)-approximable if ≥60% of its frame pairs achieved an inlier ratio ≥0.60 and a median reprojection error ≤2.0 px. Under these relatively strict thresholds, **605 out of 1,000 clips (60.5%)** are SIM(2)-approximable, with a mean inlier ratio of 0.686 and a median reprojection error of only 0.57 px.
>
> This proportion suggests that SIM(2) motion is relevant to a substantial subset of OpenVid-1M, which is also consistent with common camera and object motions in internet videos. This is expected because SIM(2) describes apparent motion on the image plane, encompassing not only camera motions (pan/tilt → translation, roll → rotation, zoom/dolly → scaling) but also any scene element whose projected motion approximates these transformations (e.g., objects approaching or receding from the camera, bulk displacement, or approximately uniform expansion/contraction). Under looser thresholds, the proportion would likely be even higher.
>
> For the remaining clips, the observability gates (Appendix A.6) and adaptive weighting (Section 3.6) limit the influence of unmatched branches, consistent with the absence of degradation on the complex-motion subset (Table 3).
>
> ## W2: Comparison with MotionCraft
>
> We agree that a comparison with MotionCraft is valuable. MotionCraft and our method differ substantially in setting: MotionCraft is a zero-shot method built on frozen Stable Diffusion v1.5 with externally simulated optical flow, whereas ours is a training-time motion prior for video diffusion backbones. Nevertheless, we performed a reference comparison for completeness.
>
> Using the officially released MotionCraft examples, we compared the two methods on MotionCraft’s reported metrics (Frame Consistency and Motion Consistency), LPIPS-flow from its supplementary code, and a standard warping error metric:
>
> |Metric|Ours|MotionCraft|
> |-|-:|-:|
> |Frame Consistency ↑|**0.9988**|0.9943|
> |Motion Consistency ↑|**0.9468**|0.8563|
> |LPIPS-flow ↓|**0.0824**|0.1250|
> |Warping Error ↓|**0.0045**|0.0104|
>
> In this reference comparison, our method performs better on all four metrics, with the largest gaps on Motion Consistency and Warping Error. If helpful, we will include this comparison and clearly state the difference in settings in the revised paper.

---

> > ### Author Rebuttal · Reviewer_CwWP · 2026-04-04
> >
> > The rebuttal addresses most of my concerns. I appreciate the statistical and experimental results and will maintain my positive score as Accept.

---

### Decision · Program_Chairs · 2026-04-30

**Decision:**

Accept (regular)

**Comment:**

This paper has mixed reviews that lean positive (1 R, 1 WA, 2 A). Reviewer praised the overall formulation of the paper (using spectral criteria to identify simple motions) and strong empirical results as well as the results on multiple backbones and the addition of a evaluation benchmark. Reviewers were concerned about the loss's implications for complex / mutliobject / fast motion as well as the thoroughness of evaluation (regarding benchmarks, comparisons against other physics-aware methods). In the rebuttal, the authors provided additional analysis on different motion types, comparison against other physics aware methods, and additional ablations as well as a Wan 2.1 data point. After the rebuttal, I advocate for acceptance.